# Factors Associated with Reclosure of Posterior Capsule Aperture by Flat Opacifications with Pearls after Nd:YAG Laser Posterior Capsulotomy

**DOI:** 10.3390/diseases11020082

**Published:** 2023-06-03

**Authors:** Akiko Ota, Ichiro Ota, Shu Kachi, Goichiro Miyake, Fuminori Haga, Kensaku Miyake, Mineo Kondo, Kumiko Kato

**Affiliations:** 1Department of Ophthalmology, Mie University Graduate School of Medicine, Tsu 5148507, Japan; mineo@med.mie-u.ac.jp (M.K.); k-kato@med.mie-u.ac.jp (K.K.); 2Shohzankai Medical Foundation, Miyake Eye Hospital, Nagoya 4620825, Japan; i_ota@me.com (I.O.); kachishu-ngy@umin.ac.jp (S.K.); goichiro.miyake@me.com (G.M.); hagafuminori@gmail.com (F.H.); miyake@spice.or.jp (K.M.)

**Keywords:** Nd:YAG laser capsulotomy, posterior capsule opacification, reclosure, re-opacification, posterior capsule aperture, Elschnig pearls, water content of IOL

## Abstract

In this retrospective case series, we investigated factors associated with posterior capsule aperture (PCA) reclosure following neodymium-yttrium aluminum garnet (Nd:YAG) laser posterior capsulotomy. The study encompassed patients who underwent cataract surgery with intraocular lens (IOL) implantation or a combined vitrectomy, cataract surgery, and IOL implantation between 2009 and 2022. PCA reclosure was observed in 22 eyes of 17 patients: 45% (10 eyes) underwent the triple procedure, and 55% (12 eyes) received cataract surgery with IOL implantation. In our clinic, 14% of patients were given IOLs with a 4% water content, while 73% (13 eyes) of those experiencing PCA reclosure had IOLs with a 4% water content. The mean interval between Nd:YAG capsulotomies was notably shorter than that between the initial cataract surgery and the first Nd:YAG laser capsulotomy. We also identified five stages of PCA reclosure progression. In conclusion, IOL water content may be linked to PCA reclosure, and the time to recurrence is shorter with each successive reclosure. Further research is needed to verify these findings and uncover additional contributing factors.

## 1. Introduction

A posterior capsular opacification (PCO), also known as aftercataract or secondary cataract, occurs in pseudophakic eyes several months to years following cataract surgery with intraocular lens (IOL) implantation in the capsular bag. PCO can result in a reduction of the best-corrected visual acuity (BCVA) [1], and neodymium-yttrium aluminum garnet (Nd:YAG) laser posterior capsulotomy is used to treat the PCO.

According to past studies, PCO was observed in 12.3% of the eyes at 6 months after the surgery and increased to 34.0% 5 years after the surgery [2]. Factors that might contribute to the likelihood of PCO development were identified as the use of hydrophilic intraocular lens (IOL) material, an axial length > 26 mm, the presence of high myopia, and the implantation of lower-power IOLs. In addition, a higher incidence of PCO was reported in patients who had prior vitrectomy, suggesting a possible link between this surgical procedure and the development of PCO. Demographic factors, such as a younger age and female sex, were also noted to be associated with an elevated risk of PCO, highlighting the relevance of individual characteristics in the onset of this condition. Notably, another study reported a higher incidence of PCO when hydrophilic IOLs had been implanted [3]. This particular finding has led to a shift in the clinical practice, with an increased usage of hydrophobic IOLs in recent cataract surgeries.

The formation of Elschnig pearls, opacities resembling a ring of pearls, around the posterior capsular aperture (PCA) has been documented in approximately 80% of cases following posterior capsulotomy for aftercataracts [4]. Although most of these cases are self-limiting, a second Nd:YAG laser capsulotomy was required in about 20% of patients [5,6]. Another research study reported that the rate of repeat Nd:YAG laser capsulotomy was 0.31% among cases that received an initial laser treatment [7]. The risk factors for PCA reclosure included systemic diseases such as diabetes and hypertension, prior vitrectomy, retinal diseases, younger age, and implanted hydrophilic IOLs [7,8,9,10,11]. However, these factors have not been conclusively confirmed due to the limited number of cases and lack of statistical analyses in these earlier studies.

The purpose of this study was to determine the processes involved in PCA reclosure and to identify the factors significantly associated with the reclosures following Nd:YAG laser treatment. To accomplish this, we examined the medical records of 22 eyes of 17 patients who had developed PCA reclosure after Nd:YAG laser treatment for a prior PCA closure.

## 2. Materials and Methods

### 2.1. Study Design and Participants

This was a retrospective study of patients from two ophthalmological institutions. We analyzed the medical records of 22 eyes of 17 patients who had a PCA reclosure after Nd:YAG laser posterior capsulotomy. These eyes had undergone either phacoemulsification cataract surgery with IOL implantation or a combined pars plana vitrectomy, phacoemulsification cataract surgery, and IOL implantation (triple procedures) at the Miyake Eye Hospital or an associated hospital between 2009 and 2022. The lens cortex was completely removed by posterior capsular polishing during the cataract surgery in all cases. Patients with PCO or PCA reclosure accompanied by reduced best-corrected visual acuity (BCVA) or those who reported visual disturbances were treated with Nd:YAG laser capsulotomy with VISULAS^®^ YAG III (ZEISS). Capsulotomy was performed by four experienced surgeons (IO, SK, GM and KM). All patients were instructed to use topical non-steroidal anti-inflammatory drugs on the treated eye 2–3 times/day for 7–14 days following the Nd:YAG laser capsulotomy.

### 2.2. Ethical Considerations

The procedures used in this study adhered to the tenets of the Declaration of Helsinki, and all patients provided informed consent for the use of their case details and accompanying images in future publications.

### 2.3. Statistical Analyses

Mann–Whitney U tests were used to assess the significance of differences in the findings between patients who had undergone cataract surgery and those who had undergone triple surgery. A two-way layout analysis of variance was performed to determine the significance of the differences in the mean interval of each Nd:YAG laser treatment. Subsequently, Dunnett’s multiple comparison tests were used to determine whether the intervals between the first and second laser treatments and the second and third laser treatments were significantly shorter than the interval between the initial cataract surgery and the first laser treatment. The decimal visual acuities were converted to the logarithm of the minimum angle of resolution (logMAR) units for statistical analyses. Data are presented as the means ± SDs, and *p* values < 0.05 were considered statistically significant. Statistical analyses were performed using BellCurve for Excel (Social Survey Research Information Co., Ltd., Tokyo, Japan).

## 3. Results

### 3.1. Clinical Characteristics and Demographics of Participants

The clinical characteristics of the patients are presented in Table 1, and their demographics are presented in Table 2. A total of 22 eyes from 17 patients (5 eyes of 5 men and 17 eyes of 12 women) had a PCA reclosure after Nd:YAG laser posterior capsulotomy. Of these, 12 eyes from 9 patients had undergone cataract surgery with IOL implantation, while 10 eyes from 9 patients had undergone the triple procedures. One patient had undergone cataract surgery in one eye and triple procedures in the other eye. The mean age of all patients at the time of surgery was 66.4 ± 13.6 years (range, 36 to 86 years). Three patients were being treated for diabetes mellitus, and nine patients for hypertension. The differences in the age, sex distribution, and systemic diseases between the cataract surgery group and the triple procedures group were not significant (Table 2). Of the 10 eyes that underwent the triple procedures, 9 eyes had an epiretinal membrane.

### 3.2. Characteristics of Intraocular Lens

The IOLs used in the cases that developed a reclosure of a PCA are shown in Table 1. The 16 (73%) eyes that developed a PCA reclosure had the Eternity^®^ IOL (Santen, Inc., Osaka, Japan) implanted. In contrast, a review of the clinical records of 23,660 eyes that underwent cataract surgery at our hospitals over the past five years revealed that only 3357 eyes (14%) received the Eternity IOL. The odds ratio for PCA reclosure associated with the use of Eternity IOL was 16.2.

The Eternity IOL possesses a water content of 4%. In contrast, other lenses which resulted in PCA reclosure consisted of five eyes with a water content of less than 1%, and a singular case with a water content of 1.5% using a Clareon^®^ IOL (Alcon Laboratories, Inc., Fort Worth, TX, USA). It should be noted that all IOLs employed in the study had a sharp optic edge.

### 3.3. Interval between Laser Capsulotomies

The mean interval between the initial cataract surgery and the first Nd:YAG laser posterior capsulotomy (YAG-1) was 19.5 ± 8.7 months (range, 4.3 to 37.3 months). The mean interval between YAG-1 laser posterior capsulotomy and the second Nd:YAG (YAG-2) capsulotomy was 10.4 ± 5.8 months (range, 4.3 to 26.6 months; Table 3). Seven eyes of 7 patients had a PCA reclosure after YAG-2 capsulotomy with a mean interval between YAG-2 and YAG-3 of 6.0 ± 1.7 months (range, 2.8 to 8.1 months). Comparing the interval between YAG-1 and YAG-2 in 22 patients with PCA reclosure after Nd:YAG laser treatment for PCO showed that YAG-2 was significantly shorter than YAG-1 (Figure 1A, *p* < 0.001). Comparing the intervals between YAG-1, YAG-2, and YAG-3 in 7 patients who underwent Nd:YAG capsulotomy across three sessions showed that both YAG-2 and YAG-3 were significantly shorter than YAG-1 (Figure 1B, *p* = 0.017 and *p* = 0.004, respectively). However, the difference in the mean intervals of laser treatment between the cataract surgery and triple procedure groups was not significant.

### 3.4. Staging of Posterior Capsule Reclosure

The sequence of events leading to PCA reclosure was classified into five distinct stages in this study. Immediately following the Nd:YAG laser posterior capsulotomy, Stage 0 is characterized by an absence of lens epithelial cells (LECs) proliferation around the posterior capsulotomy borders, leaving the visual axis clear (Figure 2A). Progressing to Stage 1, we observed pearl-like proliferations surrounding the posterior capsulotomy borders without any discernible opacities on the intraocular lens (IOL) (Figure 2B). This evolutionary process then enters Stage 2, where the development of spiculated proliferative tissue adorned with pearl-like opacities begins to extend from the capsulotomy borders toward the center of the IOL (Figure 2C). This growth suggests the initiation of a more advanced reclosure process, marked by the intertwining of these spiculated tissues. In Stage 3, these intricate tissues merge and establish bridges, which subsequently culminate in reticulated opacities (Figure 2D). The progression culminates in Stage 4, where the posterior surface of the IOL is veiled by a thin layer of opacities, filling in the gaps present in the Stage 3 reticulated opacities (Figure 2E).

We observed that the flat pearl-like opacifications had spontaneously decreased in some cases during the follow-up period, which resulted in a downstaging of the disease process.

A representative case with PCA reclosure is shown in Figure 3. After the initial Nd:YAG laser capsulotomy for a PCO following cataract surgery, a decrease in the BCVA occurred due to a PCA reclosure leading to a Stage 3 opacification on the posterior surface of the IOL at 15 months (Figure 3A). The patient then received a second Nd:YAG laser treatment (Figure 3B), and 1 month later, flat pearl-like proliferations were observed along the posterior capsulotomy border (Figure 3C). Subsequently, the proliferative tissue gradually spread to the posterior surface of the IOL, and 5 months later, the PCA was closed by flat pearl-like opacification, resulting in Stage 4 opacification (Figure 3D). The BCVA was reduced to 0.18 logMAR units. After a third laser treatment (Figure 3E), the visual acuity recovered to −0.18 logMAR units.

All 22 eyes that required Nd:YAG laser treatment for visual reduction had Stage 3 or 4 PCA reclosure, with 5 eyes at Stage 3 and 17 eyes at Stage 4. The average decrease in BCVA before the treatment for the first PCA reclosure was 0.27 ± 0.17 logMAR units (range, 0 to 0.70), and, in all cases, the average BCVA improved to −0.25 ± 0.14 logMAR units (range, −0.56 to −0.03) after the Nd:YAG laser treatment. Seven eyes that developed PCA reclosure with a decrease in BCVA even after two sessions of Nd:YAG laser treatments had Stage 3 or a higher degree of PCA reclosure. The average BCVA before treatment for the second PCA reclosure was 0.19 ± 0.09 logMAR units (range, 0.05 to 0.30), and it improved to −0.20 ± 0.16 logMAR units (range, −0.48 to 0) after the Nd:YAG laser treatment. None of the patients with Stage 2 or lower was aware of their decreased BCVA.

## 4. Discussion

We investigated 22 eyes that developed a PCA reclosure that was characterized by flat opacifications with Elschnig pearls following Nd:YAG laser capsulotomy for a PCO. We monitored the PCA reclosure process and established a five-stage classification system. A decline in the BCVA was consistently observed at Stage 3 and worse, and this classification system proved useful for determining the appropriate time for Nd:YAG laser treatment.

According to a 2022 report, the incidence of PCO following cataract surgery is relatively high with rates of 12.3% at 6 months, 4.4% at 1 year, 19.7% at 3 years, 34% at 5 years, and 46.9% at 9 years [2]. Previous studies have shown that 83% of patients that had undergone Nd:YAG laser treatment for PCO developed pearl-like proliferative tissue at the PCA border [4]. This phenomenon was suggested to result from the Nd:YAG laser capsulotomy triggering a healing response which, in turn, stimulated LEC proliferation and migration. These changes lead to the formation of the pearl-like opacities. While the majority of pearl-like proliferations are self-limiting and resolve spontaneously, certain cases involving implanted IOLs have proliferative tissue extension beyond the PCA, which ultimately results in PCA reclosure [5,6]. The time to PCA reclosure shortened with a repeated reclosure of a PCA, and this result suggests that the Nd:YAG laser treatment itself may have triggered reclosure of the PCA. To test this hypothesis, it is necessary to compare the number of shots and total energy for laser capsulotomy, but this information was missing in some cases. Thus, analysis could not be performed. Further studies will be needed to analyze the relationship between Nd:YAG laser treatment and the development of PCA reclosures.

The staging system for PCOs proposed by Congdon et al. was based on the opacity density [12]. In contrast, our classification for PCA reclosure is based on the expansion of opacities on the posterior surface of the IOL. Nd:YAG laser capsulotomy was performed only in cases at Stage 3 or 4 where the visual axis was obstructed by the pearl-like opacities. This then suggested that our PCA reclosure classification could be helpful in determining when Nd:YAG treatment is needed. We also found it interesting that the development of opacifications on the visual axis at Stage 3 or 4 could spontaneously resolve in some cases. This type of spontaneous resolution was not observed in PCOs and appears to be unique to PCA reclosures. The underlying mechanisms behind this spontaneous resolution remain unclear and warrant further investigation.

Previous studies have proposed several risk factors for PCA reclosure, including the presence of retinal diseases, e.g., diabetic retinopathy, prior vitrectomy, use of hydrophilic IOLs, and younger age [7,8,9,10,11]. In our cohort, 18% of patients had diabetes mellitus, and 52% had hypertension. Considering that the prevalence of diabetes and hypertension in Japan is approximately 20% and 50%, respectively [13], these conditions are unlikely to serve as significant risk factors for developing PCA reclosure in our study population. However, these factors should not be disregarded, and the possibility of a multifactorial etiology should be explored. Furthermore, we observed no statistically significant difference in the time to Nd:YAG laser treatment between the cataract surgery group and the triple procedure group (Table 3).

In contrast, despite the general preference for hydrophobic IOLs in our practice, the Eternity IOL had higher rates of PCA reclosure compared to other IOLs with an odds ratio of 16.2. The design of this IOL features a sharp optic edge, a characteristic also present in other IOLs used in our hospital. However, the unique attribute of this lens material is its water content. The Eternity IOL is a novel hydrophobic IOL, manufactured by combining the optimal features of conventional hydrophobic and hydrophilic polymers to create a ‘glistening-free’ lens [14,15]. Consequently, the water content of this IOL is 4%, which is higher than that of other hydrophobic IOLs. Given that a significant number of cases with PCA reclosure in our study involved this lens, it is plausible that the higher water content of 4% may contribute to the PCA reclosure. This particular aspect warrants further investigation, specifically, additional research on the impact of different IOLs and their material properties on PCA reclosure. These findings can be used to improve surgical practices and the choice of IOLs for patients with a high risk of PCO, potentially leading to lower rates of PCA reclosure and improved visual outcomes for these patients.

Given that patient-related factors such as age, the presence of systemic conditions such as diabetes and hypertension, and the type of surgical procedure performed (cataract surgery vs. triple procedure) were not associated with PCA reclosure in our study, this underscores the potential importance of surgical factors, including the choice of IOL. However, it should be noted that our findings are preliminary and further research is needed to confirm these observations and to explore potential mechanisms.

Finally, the influence of an individual’s sex on the reclosure of PCA after Nd:YAG treatment for PCO warrants investigation. In our cohort, a significant majority, comprising 70% of the cases, were women. Previous research, notably the study conducted by Congdon et al., indicated a significantly higher occurrence of PCO in women [12]. Consequently, we hypothesize that the female sex may be associated with the development of PCA reclosure following Nd:YAG laser treatment. This potential correlation highlights the necessity for clinicians to remain cognizant of sex as a possible influencing factor when managing PCO in women.

The possibility of other unknown factors contributing to PCA reclosure after Nd:YAG laser capsulotomy for PCO cannot be disregarded, and future research should strive for a comprehensive understanding of this condition. This comprehensive approach can help to inform more effective strategies for PCO management and the prevention of PCA reclosure, ultimately improving patient care and outcomes.

Our study has several limitations. First, a potential selection bias may have existed due to the longer-than-average time interval between surgery and the development of PCO. This, along with the absence of a control group, could impact the validity of our findings. Furthermore, the study requires a more substantial prospective investigation to accurately determine the risk factors for PCA reclosure following Nd:YAG treatment for PCO. Another notable limitation involves our inability to thoroughly analyze the correlation between PCA reclosure and laser capsulotomy. This gap arises from the absence of data on the number of shots and total energy used during the capsulotomy procedure. Lastly, our knowledge of the frequency of PCA reclosure post Nd: YAG laser capsulotomy is hindered by the lack of data regarding the total number of laser capsulotomies performed for PCO during the study period. Therefore, we recommend that future studies consider these factors to ensure a comprehensive analysis of PCOs and their associated treatment outcomes.

In conclusion, we studied cases with PCA reclosure after Nd:YAG laser capsulotomy for a PCO and were able to classify the reclosure process into five stages. This staging was helpful in determining the appropriate time for Nd:YAG laser treatment. Additionally, the water content of IOLs appeared to play a role in the development of PCA reclosure after Nd:YAG laser capsulotomy. Our findings contribute to the existing knowledge on PCA reclosures and their potential risk factors. Future studies are needed to determine the role of IOL water content in PCA reclosure more thoroughly as well as to explore possible interventions to mitigate this risk. Further research may also help refine the classification system for PCA reclosure and enhance our understanding of its progression, ultimately leading to improved patient outcomes and clinical management strategies.

## Figures and Tables

**Figure 1 diseases-11-00082-f001:**
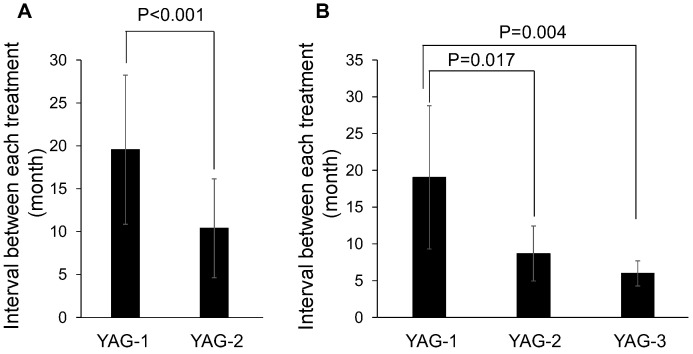
Statistical analyses of the interval between each Nd:YAG laser capsulotomy. (**A**) The interval between the first Nd:YAG laser capsulotomy and the second Nd:YAG capsulotomy (YAG-2) was significantly shorter than the mean interval between the initial cataract surgery and the first Nd:YAG laser posterior capsulotomy (YAG-1). (**B**) YAG-2 and the mean interval between the second Nd:YAG capsulotomy and the third Nd:YAG laser capsulotomy (YAG-3) was significantly shorter than YAG-1.

**Figure 2 diseases-11-00082-f002:**
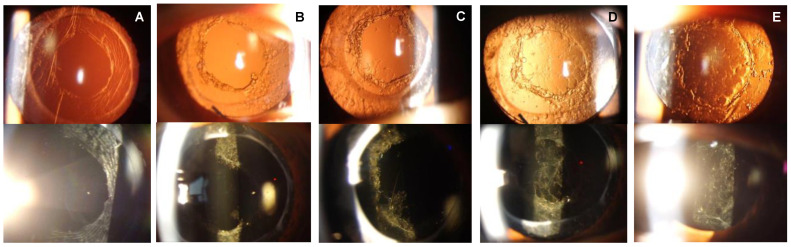
The upper panel shows images of the anterior segment taken by the retroillumination method and the lower panel by the direct illumination method. (**A**) Stage 0: Absence of lens epithelial cell (LEC) proliferation around the borders of the capsulotomy. (**B**) Stage 1: Development of Elschnig pearls around the border of the capsulotomy, but no opacifications on the intraocular lens (IOL). (**C**) Stage 2: Development spinous opacifications on the peripheral border of the IOL, but no opacification over the visual axis. (**D**) Stage 3: Development of mesh-pattern opacifications with pearl-like opacifications over the IOL on the visual axis. (**E**) Stage 4: Progression of the opacification with the presence of pearl formation and fibrosis all over the IOL.

**Figure 3 diseases-11-00082-f003:**
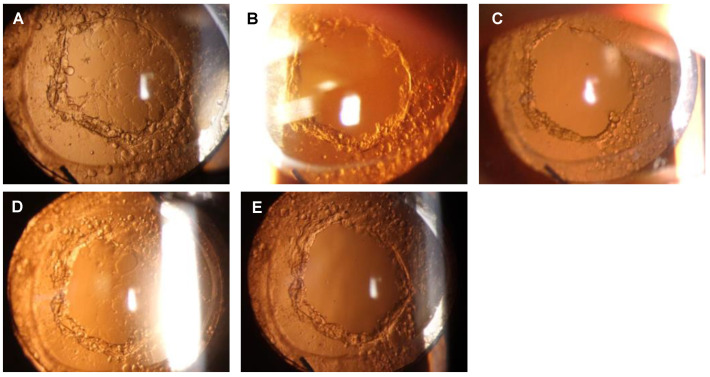
Reclosure of the posterior capsule aperture after Nd:YAG laser treatment in a representative case. (**A**) Fifteen months after the Nd:YAG laser treatment for a posterior capsule opacification (PCO). Stage 3: development of a mesh-pattern opacification with flat pearls over the IOL including on the visual axis. (**B**) Immediately after the Nd:YAG laser treatment for the reclosure of a posterior capsule aperture (PCA). (**C**) One month after the Nd:YAG laser treatment for the reclosure of PCA. Stage 1: Development of Elschnig pearls around the border of the capsulotomy but no opacifications on the IOL. (**D**) Five months after the Nd:YAG laser treatment for a PCA. Stage 4: Progression of the opacification with flat pearl formation and fibrosis all over the IOL. (**E**) Immediately after the Nd:YAG laser treatment for a closed PCA.

**Table 1 diseases-11-00082-t001:** Clinical characteristics of patients who received repeated Nd:YAG laser for reclosure of aperture in posterior capsule.

Surgery	Eye	Age	Sex	Ophthalmic Disease	Systemic Disease	IOL Model	IOL Diameter (mm)	Water Content (%)	YAG1(M)	YAG2 (M)	YAG3 (M)
PI	R	85	M	None	HTN, CI, PCa	ZA9003	6	<1.0	12.1	7.6	6.0
	L			None		ZA9003	6	<1.0	12.4	15.1	
	R	74	F	uveitis	DM, HTN	X-60	6	4	14.0	12.1	7.0
	R	59	F	None	None	NX-60	6	4	31.4	4.5	8.1
	L			None		NX-60	6	4	17.7	14.5	
	R	66	F	None	Marfan, HTN	X-70	7	4	25.6	9.1	
	L			None		X-70	7	4	25.5	4.4	
	R	39	F	None	None	SN60WF	6	0.4	37.3	6.6	
	L	76	M	None	HTN, BPH	NX-70S	7	4	27.4	4.3	
	L	76	F	None	Anemia	SY60WF	6	1.5	10.3	26.6	
	L	57	M	None	None	255	6	0.24	24.1	5.9	
	R	36	F	None	None	XY1	6	0.79	2.4	11.6	
PPV + PI	L	74	F	ERM	HTN, CI	NX-70	7	4	34.5	6.5	6.8
	R	63	F	ERM	None	NX-70	7	4	17.2	22.3	
	R	71	F	ERM, DR	HTN, DM	NX-70	7	4	24.0	6.5	
	L			ERM, DR		NX-70	7	4	23.8	6.5	
	L	66	M	ERM	None	NX-70	7	4	17.5	9.8	
	L	60	F	ERM	None	NX-70S	7	4	17.6	14.7	4.9
	R	76	M	ERM	HTN, BPH	NX-70S	7	4	20.0	11.4	
	L	61	M	ERM, glaucoma	None	NX-70S	7	4	11.5	13.3	
	R	80	F	DME	DM, HTN	NX-70S	7	4	12.3	10.0	6.3
	R	86	F	ERM	HTN	NX-70S	7	4	11.5	5.4	2.8

PPV, Pars plana vitrectomy; PI, PEA and IOL implantation; YAG1, the interval between IOL implantation and the first Nd:YAG laser posterior capsulotomy (YAG); YAG2, the interval between the first and the second YAG; YAG3, the interval between the second and the third YAG; M, month; DR, diabetic retinopathy; ERM, epi-retinal membrane; DME, diabetic macular edema; HTN, hypertension; CI, cerebral infarction; PCa, prostate cancer; DM, diabetes mellitus; BPH, benign prostatic hyperplasia.

**Table 2 diseases-11-00082-t002:** Demographic of patients.

	Overall (17 Patients)	P + I (9 Patients)	P + I + PPV (9 Patients)	*p*-Value
Age (years)	66.4 ± 14.0	63.1 ± 17.0	70.8 ± 9.0	0.376
Sex (Male/Female)	5/12	3/6	3/6	1.000
Diabetes mellitus	3	1	2	0.585
Hypertension	9	4	5	0.684

PPV, pars plana vitrectomy; P + I, PEA and IOL implantation.

**Table 3 diseases-11-00082-t003:** Interval between Nd:YAG laser capsulotomy.

	Overall (22 Eyes)	P + I (12 Eyes)	P + I + PPV (10 Eyes)	*p*-Value
YAG-1 (months)	19.6 ± 8.7	20.1 ± 10.1	19.0 ± 7.1	0.575
YAG-2 (months)	10.4 ± 5.8	10.2 ± 6.4	10.6 ± 5.2	0.717
	Overall (7 eyes)	P + I (3 eyes)	P + I + PPV (4 eyes)	*p*-value
YAG-3 (months)	6.0 ± 1.7	7.0 ± 1.1	5.2 ± 1.8	0.229

PPV, pars plana vitrectomy; P + I, PEA and IOL implantation; YAG-1, the interval between IOL implantation and the first Nd:YAG laser posterior capsulotomy (YAG); YAG2, the interval between the first and the second YAG; YAG3, the interval between the second and the third YAG.

## Data Availability

The data are available from the corresponding author upon reasonable request.

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
