# Peer review of "Factors Associated with Reclosure of Posterior Capsule Aperture by Flat Opacifications with Pearls after Nd:YAG Laser Posterior Capsulotomy"

_diseases, 2023, doi:10.3390/diseases11020082_

Round 1

Reviewer 1 Report

I read with great interest the present version of this retrospective study.

Some point should be clarified.

Pag2 lines 57-58 "routine posterior capsular polishing" should be  extensively explained,

It should be indicated the surgeon that performed capsulotomy.

It should be reported number of shots and total energy utilized for capsulotomy,

It should be specified model of Nd Yag laser utilized 

Reviewer 2 Report

This study includes a total of 22 patients who experienced PCA reclosure after undergoing intraocular lens implantation at Miyake Eye Hospital and its affiliated hospitals between 2009 and 2022. The new staging system for PCA reclosure is introduced, accompanied by clear anterior segment photographs, which are highly valuable for clinical purposes. To enhance the manuscript's quality, the following suggestions are provided:

1.    Can you calculate the frequency of PCA reclosure and provide the total number of YAG capsulotomies performed on PCOs between 2009 and 2022? If calculating this data is not feasible, please provide an explanation.

2.    It is recommended to remove lines 81-83, as they resemble instructional text and are not directly relevant to this study.

3.    In lines 85-87, the sentence "This section...can be drawn" should be omitted for the same reason stated above.

4.    In line 94, the phrase "they consisted of 5 men and 17 women" may imply "5 men and 17 women" as separate eyes rather than separate individuals. To avoid confusion with the sentence in line 89, it is suggested to combine lines 89 and 94 as follows: "(5 eyes of 5 men and 17 eyes of 12 women)." Consequently, it would be better to remove the phrase ", and they consisted of 5 men and 17 women" from line 94.

5.    There is a misspelling in line 101. "Pers" should be changed to "Pars."

6.    In line 152, please delete the letter "F" since panel F does not exist in Figure 2.

Round 2

Reviewer 1 Report

The manuscript include now suggestions previously indicated,

Reviewer 2 Report

The points suggested by the previous review have been well addressed.